# Characterisation of tetraspanins from *Schistosoma haematobium* and evaluation of their potential as novel diagnostic markers

Gebeyaw G. Mekonnen[1,2], Bemnet A. Tedla[1], Mark S. Pearson[1]*, Luke Becker[1], Matt Field[3,4,5], Abena S. Amoah[6,7,8], Govert van Dam[6], Paul L. A. M. Corstjens[9], Takafira Mduluza[10,11], Francisca Mutapi[11,12], Alex Loukas[1]*, Javier Sotillo[1,13]*

**1** Centre for Molecular Therapeutics, Australian Institute of Tropical Health and Medicine, James Cook University, Cairns, Australia, **2** Department of Medical Parasitology, School of Biomedical and Laboratory Sciences, College of Medicine and Health Sciences, University of Gondar, Gondar, Ethiopia, **3** Australian Institute of Tropical Health & Medicine and Centre for Tropical Bioinformatics and Molecular Biology, James Cook University, Cairns, Australia, **4** Immunogenomics Lab, Garvan Institute of Medical Research, Darlinghurst, Australia, **5** Menzies School of Health Research, Charles Darwin University, Darwin, Australia, **6** Department of Parasitology, Leiden University Medical Center, Leiden, The Netherlands, **7** Department of Population Health, Faculty of Epidemiology and Population Health, London School of Hygiene and Tropical Medicine, London, United Kingdom, **8** Malawi Epidemiology and Intervention Research Unit, Chilumba, Malawi, **9** Department of Cell and Chemical Biology, Leiden University Medical Center, Leiden, The Netherlands, **10** Biochemistry Department, University of Zimbabwe, P.O. Box MP167, Mount Pleasant, Harare, Zimbabwe, **11** Tackling Infections to Benefit Africa Partnership, NIHR Global Health Research Unit, University of Zimbabwe, Mount Pleasant, Harare, Zimbabwe, **12** Institute of Immunology & Infection Research, Ashworth Laboratories, University of Edinburgh, King's Buildings, Edinburgh, United Kingdom, **13** Centro Nacional de Microbiología, Instituto de Salud Carlos III, Majadahonda, Madrid, Spain

* mark.pearson@jcu.edu.au (MSP); alex.loukas@jcu.edu.au (AL); javier.sotillo@isciii.es (JS)

**Data Availability Statement:** All relevant data are within the manuscript and its Supporting Information files.

## Abstract

*Schistosoma haematobium* is the leading cause of urogenital schistosomiasis and it is recognised as a class 1 carcinogen due to the robust association of infection with bladder cancer. In schistosomes, tetraspanins (TSPs) are abundantly present in different parasite proteomes and could be potential diagnostic candidates due to their accessibility to the host immune system. The large extracellular loops of six TSPs from the secretome (including the soluble excretory/secretory products, tegument and extracellular vesicles) of *S. haematobium* (*Sh*-TSP-2, *Sh*-TSP-4, *Sh*-TSP-5, *Sh*-TSP-6, *Sh*-TSP-18 and *Sh*-TSP-23) were expressed in a bacterial expression system and polyclonal antibodies were raised to the recombinant proteins to confirm the anatomical sites of expression within the parasite. *Sh*-TSP-2, and *Sh*-TSP-18 were identified on the tegument, whereas *Sh*-TSP-4, *Sh*-TSP-5, *Sh*-TSP-6 and *Sh*-TSP-23 were identified both on the tegument and internal tissues of adult parasites. The mRNAs encoding these TSPs were differentially expressed throughout all schistosome developmental stages tested. The potential diagnostic value of three of these *Sh*-TSPs was assessed using the urine of individuals (stratified by infection intensity) from an endemic area of Zimbabwe. The three *Sh*-TSPs were the targets of urine IgG responses in all cohorts, including individuals with very low levels of infection (those positive for circulating anodic antigen but negative for eggs by microscopy). This study provides new antigen candidates to immunologically diagnose *S. haematobium* infection, and the work presented

**Funding:** This work was supported by a program grant (APP# 1037304) from the National Health and Medical Research Council (NHMRC) and a Senior Principal Research fellowship from NHMRC to AL (APP# 1117504). GGM received funding from the Australian Institute of Tropical Health and Medicine PhD scholarship. J.S. is a Miguel Servet Fellow funded by Instituto de Salud Carlos III (CP17III/00002). The funders had no role in study design, data collection, analysis and publication.

**Competing interests:** The authors have declared that no competing interests exist.

here provides compelling evidence for the use of a biomarker signature to enhance the diagnostic capability of these tetraspanins.

## Author summary

*Schistosoma haematobium*, the leading cause of urogenital schistosomiasis, affects millions of people worldwide. Infection with this parasite is associated with different clinical complications such as squamous cell carcinoma and genital malignancy in women. Despite its importance, there is a lack of sensitive and specific diagnostics that support control and elimination initiatives against this devastating disease. Herein, we have characterised six molecules belonging to the tetraspanin family of membrane proteins, providing details about their relative expression during parasite's development and their localization in adult forms of *S. haematobium*. Furthermore, we have characterised the antibody responses against three of these molecules in urine from infected human subjects from an endemic area, providing compelling evidence for the use of these molecules to diagnose urogenital schistosomiasis.

## Introduction

Schistosomiasis is a parasitic disease caused by blood dwelling trematodes from the genus *Schistosoma* and it is the second most important parasitic disease next to malaria in terms of social, economic and public health impact [1]. Over 250 million people are affected and 700 million people live in areas at risk [2]. Most of the schistosomiasis burden is found in Sub-Saharan Africa (SSA) [3], where around 280,000 people die annually [2]. *S. haematobium*, the causative agent of urogenital schistosomiasis, is endemic in the Middle East and some African countries [4, 5], and has been sporadically detected in India [6] and France [7]. The disease affects more than 90 million people, with nearly 150,000 deaths per year [8].

Pathogenesis associated with urogenital schistosomiasis is mainly caused by eggs trapped in tissues, particularly in the bladder [9]. The adult female worms live in the perivesicular veins, where a single female adult worm, when paired, can release hundreds of eggs per day after 4–7 weeks of infection [10]. Some of the eggs penetrate the bladder wall and get excreted through urine; however, half of the produced eggs are carried away with the bloodstream and trapped in the tissues of different urogenital organs [10]. The eggs trapped in the capillary beds of the bladder stimulate an immune response [9], affecting the bladder wall, leading to general urinary dysfunction and progressing to obstructive renal pathology [11, 12]. Infected individuals can also develop squamous cell carcinoma (an especially aggressive type of bladder cancer) [13, 14]; indeed, the International Agency for Research on Cancer has classified this parasite as a class 1 carcinogen [15]. In female patients, urogenital schistosomiasis may increase the risk of acquiring HIV infection [16].

Microscopic examination of urine to detect schistosome eggs is the gold standard test for diagnosis of *S. haematobium* infection [17]. Even though this technique is simple to perform and inexpensive [18], it cannot detect the acute stage of the disease (early infection, when no eggs are released) and it is influenced by day-to-day variability in egg excretion [19]. The circulating cathodic antigen (CCA) rapid diagnostic test is more sensitive than egg microscopy and detects the presence of a schistosome glycan in patient's urine [20]; however, the sensitivity of this test is low in areas where *S. mansoni* and *S. haematobium* are co-endemic [21]. A different

schistosome glycan, the circulating anodic antigen (CAA) has also be used to diagnose *S. haematobium* infections in the urine samples in low endemic settings [22, 23]. The CAA test applies a luminescent highly sensitive up-converting reporter particle technology (UCP) in a lateral flow (LF) based assay format. The UCP-LF CAA test is also applicable to blood-based samples and detects all *Schistosoma* species (potentially down to the level of a single worm [24]). However, it requires a basic laboratory and can be costly when requiring highest sensitivity; the test (yet) has not been commercialised. Molecular techniques like PCR are highly specific; however, DNA-based diagnosis of helminths requires expensive equipment and reagents [25]. Detecting antibodies produced against the different developmental stages of *S. haematobium* can help in the diagnosis of patients with a light egg load (or acute infection when the microscopic examination is still negative) in low-level endemic areas [26]. Despite the fact that antibody detection cannot differentiate between past and active infections [5, 10, 27] it could help in determining re-emergence of schistosomiasis in certain areas and the diagnosis of travellers [28] as well as supporting schistosomiasis control and elimination initiatives, with particular focus on post-elimination surveillance [29].

Recently, we documented the proteomic composition of the soluble excretory/secretory (ES) products, extracellular vesicles (EVs) and tegument from *S. haematobium* adult worms, revealing many different proteins, including tetraspanins (TSPs) [30, 31]. TSPs are a family of proteins that consist of four transmembrane domains, a small extracellular loop (SEL) and a large extracellular loop (LEL). TSPs are involved in many cellular activities such as differentiation, adhesion and division [32]. In platyhelminths, TSPs play an important role in tegument formation, maturation and stability [33–35]. In schistosomes, TSP LELs have been tested as vaccine candidates [36]; indeed, *Sm*-TSP-2 from *S. mansoni* has completed phase I clinical trials [37]. Furthermore, TSPs from other platyhelminths such as *Taenia solium* and *Schistosoma japonicum* have been suggested as potential diagnostic candidates [38, 39] and a secretome-wide immunomics/proteomics analysis of *S. haematobium* has highlighted the diagnostic efficacy of several of these molecules [29].

Herein, we characterise six *S. haematobium* TSPs (*Sh*-TSPs) that might be playing key roles in host-parasite interactions since they were found in the tegument, ES products and EVs of adult *S. haematobium* worms. We also assess the diagnostic efficacy of some of these TSPs in recombinant form using the urine of individuals with different *S. haematobium* infection intensities. This study not only provides important molecular information about this family of proteins and implicates them in host-parasite interactions, but also contributes to the first steps towards the generation of new diagnostic tools for this devastating disease.

## Materials and methods

### Ethics statement

The collection of urine from individuals from Zimbabwe was approved by the Medical Research Council of Zimbabwe; Approval MRCZ/A/1710. Written informed consent was required and obtained.

All experimental procedures performed on animals in this study were approved by the James Cook University (JCU) animal ethics committee through project A2391. All experiments were performed in accordance with the 2007 Australian Code of Practice for the Care and Use of Animals for Scientific Purposes and the 2001 Queensland Animal Care and Protection Act.

### Animals

Male BALB/c mice (6–8 week) were purchased from the Animal Resource Centre, Perth, Australia and maintained at the AITHM animal facilities on the JCU Cairns campus. Mice were

kept in cages under controlled temperature and light with free access to pelleted food and water as described previously [40].

## Human urine samples

A total of 73 urine samples from *S. haematobium* infected individuals from an endemic area and 9 urine samples from Australian volunteer donors that had never travelled to schistosomiasis endemic areas (non-endemic negative) were collected as described previously [31]. In line with WHO criteria, the infection level of urine samples collected from the endemic area were classified as either high (>50 eggs/10 mL of urine) (n = 20), medium (11–49 eggs/10 mL of urine) (n = 20), low (0.3–10 eggs/10 mL of urine) (n = 30) and egg negative (0 eggs/10 mL of urine) (n = 3). To confirm the presence or absence of infection, egg negative urines obtained from patients from the endemic area were also tested for the presence of CAA using the UCAA2000 format of the UCP-LF CAA test as described previously [41]. Of these samples, all were positive for CAA. The collected urine samples were aliquoted and placed at -80˚C until further use.

## *Schistosoma haematobium* material

*S. haematobium*-infected (Egyptian strain) *Bulinus truncatus* snails were provided by the Biomedical Research Institute, MD, USA. Snails were removed from the tank with a pair of forceps and washed several times with water to remove debris and rotifers, then transferred to a Petri dish and incubated without water at 27˚C in the dark for 2 h. Water was added and the snails were placed under light for 1.5 h at 28–30˚C. Cercariae were concentrated using a 20 μm pore size sieve and then used to infect BALB/c mice (1,000/mouse) by tail penetration [42]. *S. haematobium* adult worms were obtained by perfusion of mice at 16 weeks post-infection [42]. Freshly perfused *S. haematobium* adult worms were fixed in paraformaldehyde, embedded in paraffin and cryostatically sectioned into 7.0 μm sections [43].

   *S. haematobium* schistosomula were obtained by transforming cercariae *in vitro* by mechanical transformation as described previously [42]. Schistosomula were resuspended to a density of 1,000/ml in modified Basch media supplemented with 4x antibiotic/antimycotic (10,000 units/mL of penicillin, 10,000 μg/mL of streptomycin, and 25 μg/mL of amphotericin B) (Thermo Fisher Scientific, USA) and incubated at 37˚C in 5% $CO_2$. Media was changed daily and schistosomula (1,000) were collected at 24 hours, 3 and 5 days and immediately stored in 500 μl of TRI reagent (Sigma-Aldrich, USA) at -80˚C until further use.

## RNA extraction, cDNA synthesis and RT-qPCR

Total RNA from *S. haematobium* from 24 h, 3 d and 5 d schistosomula was extracted using TRI reagent as per the manufacturer's instruction. The RNA pellets were finally resuspended in 12 μl RNAse-free water and incubated for 5 min at 55˚C. First strand cDNA was synthesized using the total volume of the RNA suspension and Superscript III reverse transcriptase (ThermoFisher Scientific, USA) according to the manufacturer's instructions. The cDNA of *S. haematobium* adult, miracidia, cercariae and egg stages were provided by the Biomedical Research Institute, MD, USA.

   Real time quantitative PCR (RT-qPCR) was performed to check the expression levels of *Sh-tsp* genes in different life stages (adult, egg, miracidia, cercaria and schistosomula (24 h, 3 days and 5 days)). Each qRT-PCR reaction consisted of 5 μl of SYBR green premix EX Taq (2x) (Qiagen, Netherlands), 1 μl (10 mM) of each forward and reverse primer (S1 Table), 1μl (50 ng) of the first–strand cDNA and sterile water to a final volume of 10 μl. The reactions were performed on a Rotor-Gene Q (Qiagen, Netherlands) using the following conditions: initial

denaturation at 95˚C for 10 min (1 cycle) followed by 40 cycles of denaturation (95˚C for 10 sec), annealing (50˚C for 15 sec) and extension (72˚C for 20 sec). *Sh-tsp* expression was normalised to a housekeeping gene (*α-tubulin*, accession number XM_012938434.1) as described before [44] and relative expression levels were calculated using the $2^{-\Delta\Delta Ct}$ method using egg as a reference group [45].

## Phylogenetic analysis

A phylogenetic analysis was performed on the 6 *Sh*-TSPs of interest and 32 TSP sequences from different organisms (*Bos taurus*, *Clonorchis sinensis*, *Danio rerio*, *Homo sapiens*, *Mus musculus*, *S. japonicum*, *S. mansoni* and *Opisthorchis viverrini*) as well as from other TSPs present in the genome of *S. haematobium* obtained from the NCBI database (the list and accession numbers of all sequences are detailed in S2 Table). A multiple sequence alignment was carried out using the alignment program MUSCLE [46]. PhyML (v 20160207) [47] was used for maximum-likelihood (ML) phylogenetic analyses of the amino acid sequences using default parameters (substitution model: 'LG'; number of relative substitution categories: '4'; alpha value for the discrete Gamma model: 'e', which gets the maximum likelihood estimate; bootstrap values obtained using 'approximate Bayes branch support'). The tree was visualized with The Interactive Tree of Life (iTOF) online phylogeny tool (https://itol.embl.de/) [48] using default parameters.

## Cloning of *Sh-tsp* cDNAs

The cDNAs encoding for the open reading frames of 5 *Sh*-TSPs (MS3_09198, MS3_01370, MS3_01153, MS3_05226 and MS3_05289) were obtained from the *S. haematobium* genome (www.parasite.wormbase.org), while the cDNA encoding for the open reading frame of *Sh*-TSP-2 was obtained from GenBank (accession number MK238557). For ease of reading and to standardise the nomenclature with *S. mansoni* tetraspanins, we have renamed the proteins (see S3 Table for matching codes and names). The LEL regions from the 6 *Sh*-TSPs were identified using TMpred (https://embnet.vital-it.ch/software/TMPRED_form.html) and amplified by PCR using oligonucleotide primers flanking these regions and *S. haematobium* adult cDNA as a template (S4 Table). The PCR products of *Sh*-TSP-4, *Sh*- TSP-5, *Sh*-TSP-6, *Sh*-TSP-18 and *Sh*-TSP-23 were N*co*I/*Xho*I cloned into pET-32aΔHis such that they were in frame with the N-terminal thioredoxin (TRX) and C-terminal 6xHis tags. The vector pET-32aΔHis is an in-house modified version of pET-32a (Novagen, USA) which has the N-terminal 6xHis-tag absent to facilitate efficient purification after cleavage of the TRX tag if such removal is deemed necessary. The *Sh*-TSP-2 PCR product was N*de*I/*Xho*I cloned into pET41a (Novagen, USA) to facilitate native N-terminal expression without the GST fusion tag but retaining the C-terminal 6xHis tag. Recombinant vectors were transformed into *E. coli* TOP10 strains (ThermoFisher Scientific, USA) and recombination confirmed by sequencing.

## Protein expression

Recombinant plasmids were transformed into *E. coli* BL21(DE3) (ThermoFisher Scientific, USA) and resultant colonies were inoculated into 10 mL of Luria broth containing 100 μg/ml ampicillin (LB$_{amp}$) and incubated overnight at 37˚C with shaking at 200 rpm. Overnight culture was seeded (1/100) into 500 mL of fresh LB$_{amp}$ and incubated at 37˚C with shaking at 200 rpm until OD$_{600}$ = 0.5–1.0 (approximately 3 h), whereupon expression was induced by the addition of 1 mM isopropyl β-D-1-thiogalactopyranoside (IPTG) (Bioline, UK). Cultures continued incubating for 4 h (*Sh*-TSP-4, *Sh*-TSP-5, *Sh*-TSP-6 and *Sh*-TSP-23) or 16 h (*Sh*-TSP-18) and were then harvested by centrifugation at 8,000 *g* for 20 min at 4˚C. *Sh*-TSP-2 was

expressed as for *Sh*-TSP-23 except that LB supplemented with 50 μg/ml kanamycin was used instead of LB$_{amp}$.

Each harvested pellet was resuspended in 50 mL of lysis buffer (50 mM sodium phosphate pH 8, 40 mM imidazole and 300 mM NaCl), freezed/thawed 3 times and then sonicated 10 times (5 sec bursts) at 4˚C. Then, for the soluble proteins (*Sh*-TSP-2, *Sh*-TSP-6 and *Sh*-TSP-23 –determined by a small-scale pilot expression) the bacterial lysate was centrifuged at 20,000 *g* for 20 min at 4˚C and the supernatant decanted and stored at -80˚C. In the case of insoluble proteins (*Sh*-TSP-4, *Sh*-TSP-5 and *Sh*-TSP-18 –determined by small-scale pilot expression), Triton X-100 was added to a final concentration of 3% after sonication, the mixture incubated for 1 h at 4˚C with gentle shaking and then pelleted at 20,000 *g* for 20 min at 4˚C. The supernatant was removed, the pellet washed twice with 30 mL of lysis buffer (with centrifugation at 20,000 *g* for 20 min at 4˚C after each wash) and the final pellet resuspended in 20 mL of solubilisation buffer (50 mM sodium phosphate, 40 mM imidazole, 300 mM NaCl and 6 M urea). The resuspension was incubated at 4˚C overnight with gentle shaking, centrifuged at 20,000 *g* for 20 min at 4˚C and the supernatant decanted and stored at -80˚C.

An empty pET-32aΔHis vector was transformed into *E. coli* BL21(DE3) and expressed and purified as described above for *Sh*-TSP-23 to obtain the TRX tag, which was used as a control in ELISAs.

## Protein purification

Recombinant proteins were purified by Ni$^{2+}$ IMAC using an AKTA Prime UPC FPLC (GE Healthcare, USA). Each recombinant protein solution was diluted 1:4 in lysis buffer (soluble proteins) or solubilization buffer (insoluble proteins) and filtered through a 0.45 μm filter. The solutions were then applied to a 1 mL His-Trap IMAC column (GE Healthcare, USA), equilibrated with lysis buffer (soluble proteins) or solubilization buffer (insoluble proteins), at a flow rate of 1 mL/min. Bound proteins were washed with 10 column volumes (CV) of lysis buffer (soluble proteins) or solubilization buffer (insoluble proteins) and then eluted using lysis buffer (soluble proteins) or solubilization buffer (insoluble proteins) with an increasing linear gradient of imidazole (100–500 mM). Fractions containing purified recombinant proteins were combined and buffer exchanged into PBS (soluble proteins) or 1x PBS, 300 mM NaCl and 6 M urea (insoluble proteins) using a 3 kDa MWCO Amicon Ultra-15 centrifugal filter. The identity of expressed proteins was confirmed by SDS-PAGE and Western blot using anti-His monoclonal antibodies.

## Polyclonal antibody production

Three male BALB/c mice (6 weeks old) were immunized subcutaneously with 50 μg of each recombinant protein emulsified with Alum adjuvant (Thermo Fisher Scientific, USA) and boosted twice at two weekly intervals using the same amount of protein as described previously [43]. Blood was collected from each mouse before immunization and two weeks after the final immunization. Sera was collected by allowing the blood to clot followed by centrifugation at 10,000 *g* for 10 min and then stored at -20˚C.

## Immunohistochemistry

Immunohistochemistry was performed to determine the anatomic sites of *Sh*-TSP expression in sections of adult worms. Adult worm sections from *S. haematobium* were de-paraffinized using 2 x 3 min washes each of 100% and 50% xylene and rehydrated in an ethanol series. Antigen retrieval was performed by boiling the slides in citrate buffer (10 mM sodium citrate, pH 6) for 40 minutes followed by Tris buffer (10mM Tris, 1 mM EDTA, 0.05% Tween, pH 9.0),

for 20 minutes. Subsequently, sections were blocked with 10% goat serum for 1 hour at RT. After washing 3 times with TBS/0.05% Tween-20 (TBST), sections were incubated with anti-*Sh*-TSP antisera (diluted 1:50 in 1% BSA/TBST) overnight at 4˚C and then washed with TBST (3 x 5 min). Sections were finally probed with goat-anti-mouse IgG-Alexa Fluor 647 (Sigma-Aldrich, USA) (diluted 1:200 in 1% BSA/TBST) for 1 h in the dark at RT. After a final washing step with TBST, slides were mounted with Entellan mounting medium (Millipore, Germany) and covered with coverslips. The images were acquired by Nuance software with an AxioImager M1 fluorescence microscope (ZEISS, Germany).

### Indirect enzyme-linked immunosorbent assay

An enzyme-linked immunosorbent assay (ELISA) was performed to assess the diagnostic efficacy of the recombinantly expressed *Sh*-TSPs using the serum of infected mice (a pool of four independent *S. haematobium* infections and another pool of four independent *S. mansoni* infections) and the urine from naturally infected Zimbabwean people. Sera from naïve mice or urine from non-endemic individuals were used as negative controls. Microtiter plate wells (Greiner Bio-One, Austria) were coated with 50 µl (2 µg/ml) of protein (either *Sh*-TSP2, *Sh*-TSP-4, *Sh*-TSP-5, *Sh*-TSP-6, *Sh*-TSP-18 or *Sh*-TSP-23 for the mouse ELISAs and either *Sh*-TSP-4, *Sh*-TSP-5, *Sh*-TSP-18 or a combination of the three antigens for the human ELISAs) in 0.1 M carbonate-bicarbonate buffer (pH 9.6) and incubated overnight at 4˚C. The plates were washed 3 times with PBST and blocked with 5% milk at 37˚C for 1 h. Then, plates were washed 3 times with PBST and 50 µl of human urine (diluted 1:50 in PBST) or mouse serum (diluted 1:5,000 in PBST) was added and incubated at 37˚C for 1 h. After washing with PBST 3 times, 50 µl of HRP-conjugated anti-human IgG (Sigma-Aldrich, USA) or anti-mouse IgG (diluted 1:5,000 in PBST) was added, incubated at 37˚C for 1 h and washed 3 times with PBST. Finally, 50 µl of 3,3',5,5'-tetramethylbenzidine (TMB) (Thermo Fisher Scientific, USA) was added and incubated for 12 minutes at RT in the dark. The reaction was stopped with 3 M HCl and the absorbance was measured at 450 nm using a POLARstar Omega (BMG Labtech, Australia).

### Statistics

All statistics were performed using GraphPad Prism 9.0. For both serum and urine ELISAs, significance between groups was determined using the Student's t test and reactivity cut-off values were determined as the average reactivity +3SD of the non-endemic negative group. Receiver Operating Characteristic (ROC) curves were used to calculate sensitivity, specificity and the area under the curve (AUC).

## Results

### General characteristics of *Sh*-TSPs

Six *Sh*-TSPs (*Sh*-TSP-2, *Sh*-TSP-4, *Sh*-TSP-5, *Sh*-TSP-6, *Sh*-TSP-18 and *Sh*-TSP-23) were selected based on previous proteomic analyses from the adult worm tegument, ES and EVs (Table 1) [31]. The sequences of all *Sh*-TSPs were interrogated and each ORF contained four transmembrane domains, one SEL, one LEL and three intracellular regions. Each LEL contained the four-cysteine residues (forming two disulfide bonds) as well as the CCG motif characteristic of TSPs [49]. Full-length cDNA sequence, predicted amino acid sequence and LEL sizes are shown in Table 1. The amino acid sequence similarities between the *Sh*-TSPs described here and their *S. mansoni* homologs ranged from 71–93% (when entire ORFs were compared) (S1 Fig) and 70.2–84% (when just the LEL regions were compared) (S2 Fig).

**Table 1. Characteristics of *Schistosoma haematobium* tetraspanins.** Aas: aminoacids; bp: base pairs; LEL: large extracellular loop; SEL: small extracellular loop.

| TSPs | Length (bp) | Size (# aas) | Weight (kDa) | SEL position | Inner loop position | LEL position | Cytoplasmic tail | Originally identified from [30,31] |
|---|---|---|---|---|---|---|---|---|
| *Sh*-TSP-2 | 660 | 219 | 24.4 | 31–79 | 74–80 | 102–183 | 213–219 | Tegument 120k EVs 15k EVs |
| *Sh*-TSP-4 | 837 | 278 | 30.9 | 36–76 | 100–108 | 130–250 | 272–278 | Tegument 120k EVs 15k EVs |
| *Sh*-TSP-5 | 822 | 273 | 30.4 | 31–79 | 97–109 | 173–239 | 265–273 | Secretome 120k EVs 15k EVs |
| *Sh*-TSP-6 | 675 | 224 | 24.9 | 36–54 | 87–95 | 108–190 | 212–225 | Tegument 15k EVs |
| *Sh*-TSP-18 | 888 | 295 | 32.9 | 32–66 | 87–95 | 118–266 | 283–225 | 120k EVs 15k EVs |
| *Sh*-TSP-23 | 654 | 218 | 24.2 | 36–56 | 72–81 | 103–184 | 205–218 | Tegument 120k EVs 15k EVs |

## Phylogeny of *Sh*-TSPs

A phylogenetic analysis between the studied *Sh*-TSPs and other well-characterised TSPs from related trematodes was performed. *Sh*-TSP-2, *Sh*-TSP-6 and *Sh*-TSP-23 grouped together in the CD63 clade of TSPs and clustered together with other well characterised CD63-like TSPs from *S. mansoni* (AAN17276, AAA73525 and XP_018650438), *S. japonicum* (CAX70616) and *O. viverrini* (JQ678707.1 and JQ678708.1) and bootstrapping values > 0.7 (Fig 1). *Sh*-TSP-4, *Sh*-TSP-5 and *Sh*-TSP-18 clustered under the uroplakin family of TSPs together with other TSPs from *S. mansoni* (XP_018649476, XP_18653608, XP_002577444 and XP_002575497), *S. japonicum* (AAW26298, AAW26326, AAW24822, AAP05954 and AAW27174) and *S. haematobium* (MS3_02232, MS3_07569, MS3_08458, MS3_01905, MS3_03452, MS3_03944, MS3_01557, MS3_01094, MS3_03883) and bootstrap values mostly > 0.9 (Fig 1).

## Protein expression and purification

The LEL region from each of the *Sh-tsp* cDNA was cloned into a bacterial expression system and sequences were validated by Sanger sequencing. The *Sh*-TSP-6, *Sh*-TSP-23 and *Sh*-TSP-2 LELs were expressed as soluble proteins, while *Sh*-TSP-4, *Sh*-TSP-5 and *Sh*-TSP-18 were expressed as inclusion bodies and 6 M urea was used for solubilisation. The expected sizes of the expressed *Sh*-TSPs were confirmed by SDS-PAGE (S3A Fig) and a western-blot using a monoclonal anti-His antibody (S3B Fig).

## *Sh*-TSPs are expressed throughout all life stages

The transcriptional patterns of all *Sh-tsp* mRNAs were analysed in different life stages of *S. haematobium*: adult, egg, miracidia, cercaria and schistosomula (24 h, 3 d and 5 d) by RT-qPCR. All *Sh-tsp* mRNAs were expressed throughout all life stages tested (Fig 2). Overall, the expression levels of *Sh-tsp-2* and *Sh*-TSP-18 were the highest (Fig 2A and 2D). *Sh-tsp-2* expression peaked at 24 h schistosomula and decreased in subsequent developmental stages (Fig 2A). Similarly, the highest expression level of *Sh-tsp*-5, *Sh-tsp*-6 and *Sh-tsp*-18 was identified in the miracidia while the lowest level of expression was observed in adult, 3-day schistosomula and 24 h schistosomula, respectively (Fig 2B, 2F and 2D). In the case of *Sh-tsp*-23, the highest expression level was observed in cercariae and the lowest expression level was observed in

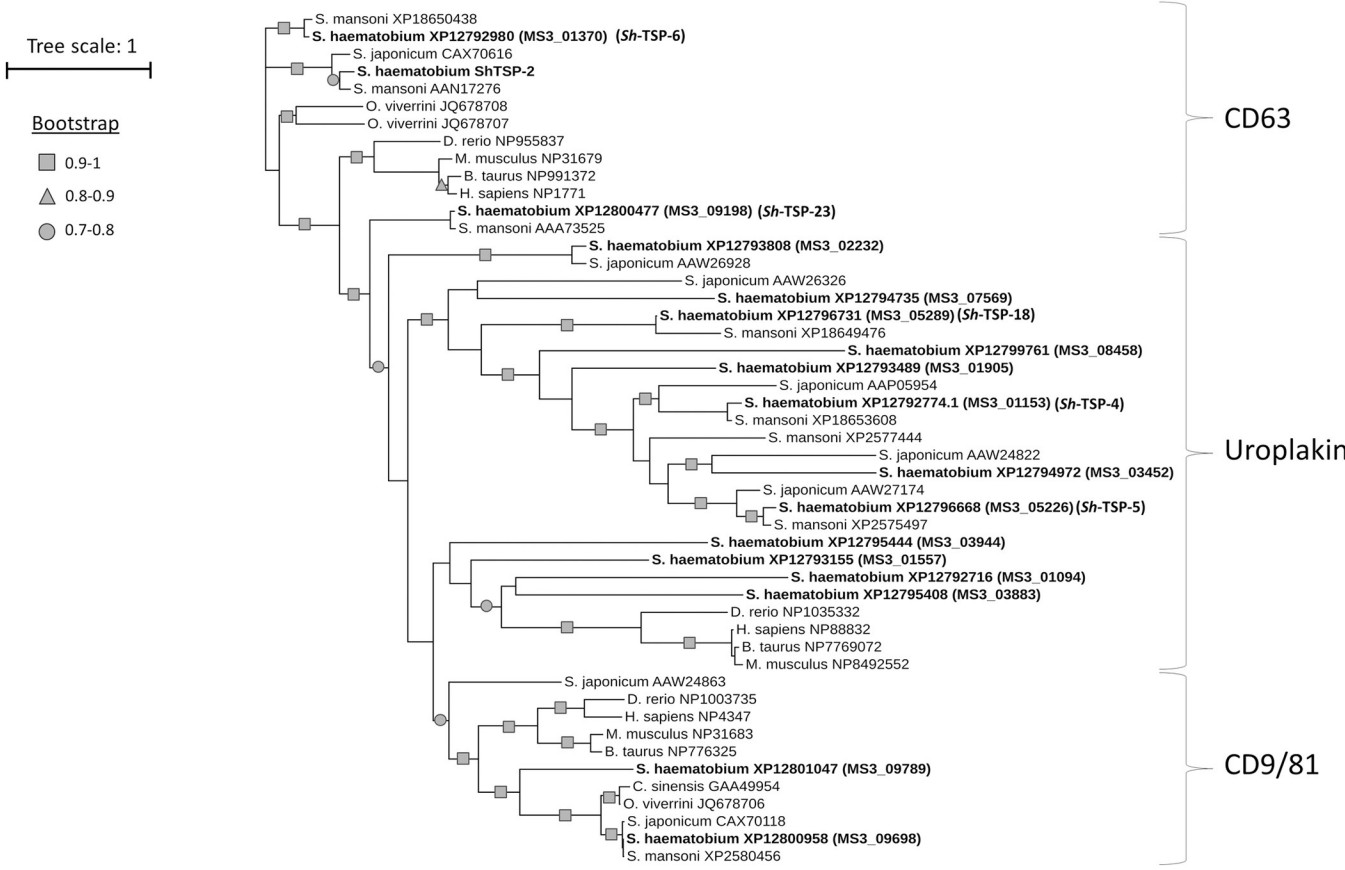

**Fig 1. Phylogenetic analysis of *Schistosoma haematobium* tetraspanins and homologs from related organisms.** A multiple sequence alignment was carried out using MUSCLE. PhyML was used for a maximum-likelihood phylogenetic analysis with bootstrapping, and results were visualized with The Interactive Tree of Life (iTOF) online phylogeny tool (https://itol.embl.de/) using default parameters.

miracidia (Fig 2C). The expression level of *Sh-tsp*-4 was highest in cercariae and lowest in adult life stages (Fig 2E).

## *Sh*-TSPs are expressed in the tegument and internal organs of *S. haematobium* adult worms

To determine the location of *Sh*-TSPs in the adult worms, sections from *S. haematobium* adult worms were probed with mouse polyclonal antibodies produced against each *Sh*-TSP (Fig 3). *Sh*-TSP-2 and *Sh*-TSP-18 were identified on the tegument of the worms, whereas *Sh*-TSP-4, *Sh*-TSP-5, *Sh*-TSP-6 and *Sh*-TSP-23 were identified both on the tegument and internal organs of adult worms (Fig 3). *S. haematobium* adult worm sections were not recognised by the negative control anti-TRX antibody (Fig 3).

## *Sh*-TSPs are recognized by the sera of infected mice

As a first step towards the assessment of TSPs as diagnostic candidates, we performed an indirect ELISA to analyse the immunogenicity of these TSPs in mice. All *Sh*-TSPs except *Sh*-TSP-18 were the target of significantly greater antibody responses ($P<0.001$) in the sera of mice experimentally infected with *S. haematobium* compared to uninfected mouse sera (Fig 4A). To assess cross-reactivity with *S. mansoni*, an indirect ELISA was performed using the sera of

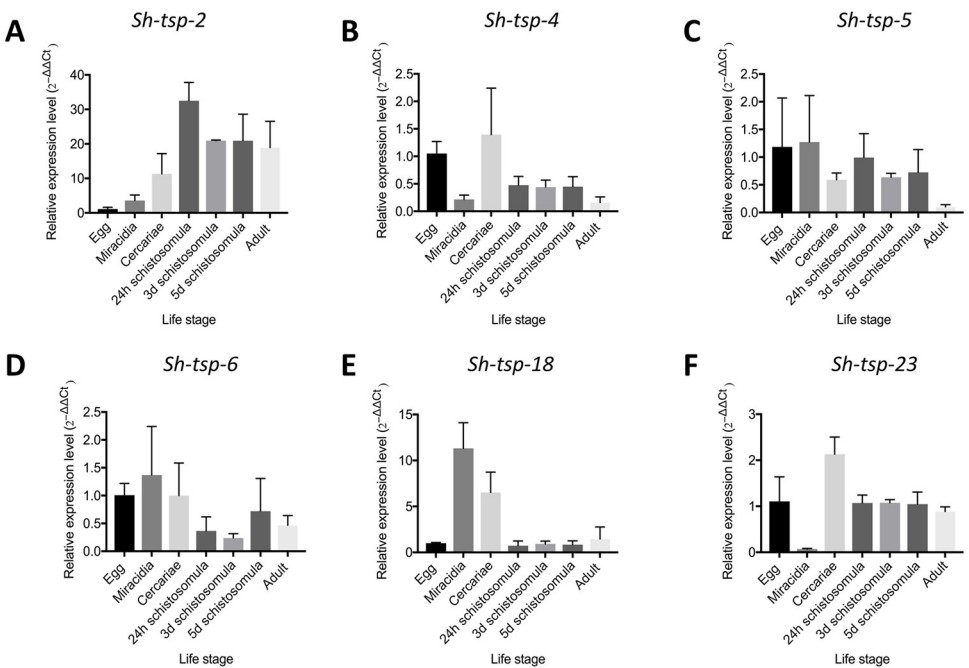

**Fig 2. Expression levels of *Schistosoma haematobium* tetraspanins at different life stages.** Relative mRNA expression levels of *Sh-tsp-2* (A), *Sh*-tsp-4 (B), *Sh*-tsp-5 (C), *Sh*-tsp-6 (D), *Sh*-tsp-18 (E) and *Sh*-tsp-23 (F) were analyzed by qPCR. Relative mRNA expression levels were normalised to a housekeeping gene ($\alpha$-tubulin), calculated using the $2^{-\Delta\Delta Ct}$ method using the egg stage as a reference group.

mice experimentally infected with *S. mansoni*, and *Sh*-TSP-4, *Sh*-TSP-5, *Sh*-TSP-6 and *Sh*-TSP-23 were the target of significantly greater ($P<0.001$) antibody levels in *S. mansoni* infected mice compared to uninfected mice (Fig 4B).

## *Sh*-TSPs are recognized by antibodies in the urine of naturally infected human subjects from an endemic area

Since the diagnostic efficacy of *Sh*-TSP-2, *Sh*-TSP-6 and *Sh*-TSP-23 has already been evaluated in a previous publication [29], herein we tested the diagnostic efficacy of the uroplakin members *Sh*-TSP-4, *Sh*-TSP-5 and *Sh*-TSP-18 via their recognition by antibodies in urine from infected individuals from an endemic area in Zimbabwe. All *Sh*-TSPs tested were the targets of significantly higher antibody levels in infected individuals compared to negative controls (urine samples from a non-endemic area), either using individual TSPs (Fig 5A, 5B and 5C) or a combination of all TSPs (Fig 5D). Since all tested proteins were expressed as thioredoxin-fused molecules, the thioredoxin tag obtained from the empty vector was used as a control. Only four sera from individuals with a high intensity infection cross-reacted with this tag, whereas thioredoxin was not recognized by other infected individuals (S4 Fig). Frequency of recognition (FoR) was above 50% for every single antigen (52% for *Sh*-TSP-4, 79% for *Sh*-TSP-5 *and* 55% for *Sh*-TSP-18,), and 88% when the three antigens were combined (Fig 5E). No correlation was observed between egg counts and OD for any of the proteins tested.

## Predictive accuracy of *Sh*-TSPs for the diagnosis of *Schistosoma haematobium* infection

The diagnostic accuracy of *Sh*-TSPs was measured by calculating the area under the curve (AUC) of the receiver operating characteristic (ROC) curve generated for each antigen (Fig 6).

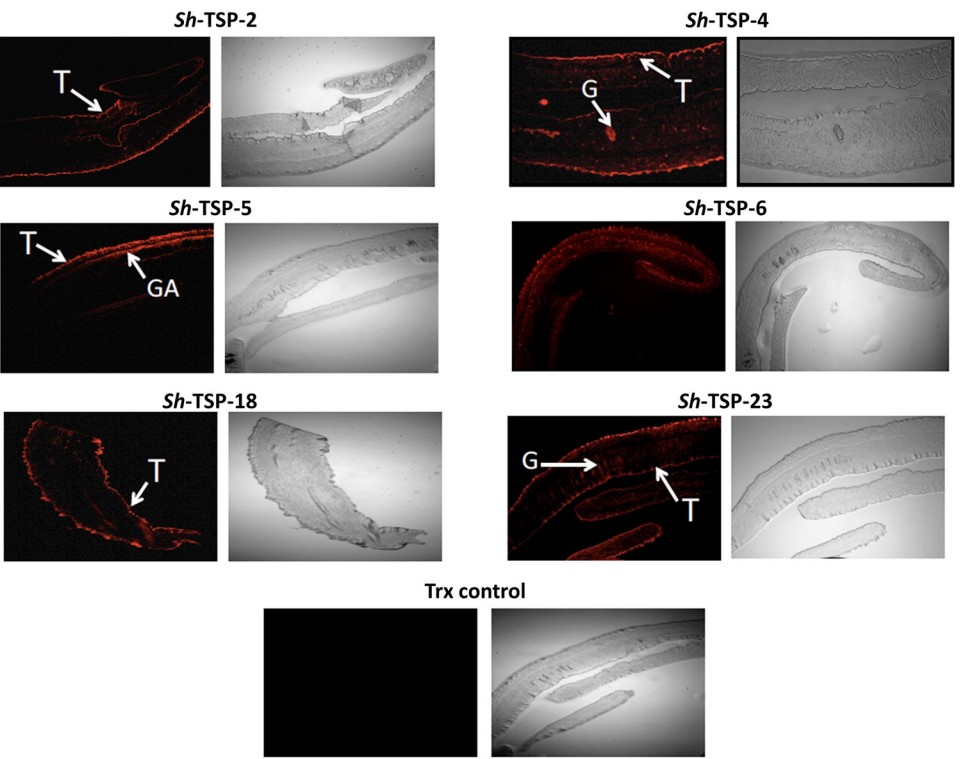

**Fig 3. Localisation of *Schistosoma haematobium* tetraspanins in adult worms.** Immunolocalisation of the *S. haematobium* tetraspanins (TSPs) *Sh*-TSP-2, *Sh*-TSP-4, *Sh*-TSP-5, *Sh*-TSP-6, *Sh*-TSP-18, *Sh*-TSP-23 and thioredoxin (TRX) in adult worm sections. Sections were probed with anti-*S. haematobium* TSP sera followed by goat-anti-mouse IgG-Alexa Fluor 647.

All three antigens showed a diagnostic accuracy of 0.99 for individuals with high egg burdens (Fig 6). In the case of medium egg burden individuals, the highest accuracy of diagnosis was obtained with *Sh*-TSP-5 (0.97), followed by *Sh*-TSP-18 (0.91) and *Sh*-TSP-4 (0.87) (Fig 6). For individuals with low egg burdens, the highest accuracy of detection was also obtained with *Sh*-TSP-5 (0.88), followed by *Sh*-TSP-18 (0.82) and *Sh*-TSP-4 (0.81) (Fig 6). In the case of egg negative but CAA positive individuals, the highest accuracy of detection was obtained with *Sh*-TSP-5 (0.72) followed by *Sh*-TSP-4 and *Sh*-TSP-18 (both with 0.70) (Fig 6).

The combination of *Sh*-TSP-4, *Sh*-TSP-5 and *Sh*-TSP-18 resulted in higher AUCs for all cohorts. An AUC of 1.0 was obtained in the case of high and medium egg burden individuals, while in the case of low and egg negative/CAA positive burden individuals, accuracy of detection was 0.90 and 0.70, respectively.

Additionally, the AUC for all infected was 0.95 when using the combination of *Sh*-TSP-4, *Sh*-TSP-5 and *Sh*-TSP-18 antigens, while it was 0.87, 0.93, 0.88 and for *Sh*-TSP-4, *Sh*-TSP-5 and *Sh*-TSP-18, respectively.

## Discussion

TSPs are a family of proteins that consist of four transmembrane domains, a SEL and a LEL. The LEL portion of TSPs contains between four and eight cysteine residues that form two to four disulfide bonds, facilitating specific protein-protein interactions with laterally associated proteins and other ligands [50]. TSPs are involved in many activities of cells such as differentiation, adhesion, and division [32]. They can also play an important role in host-parasite

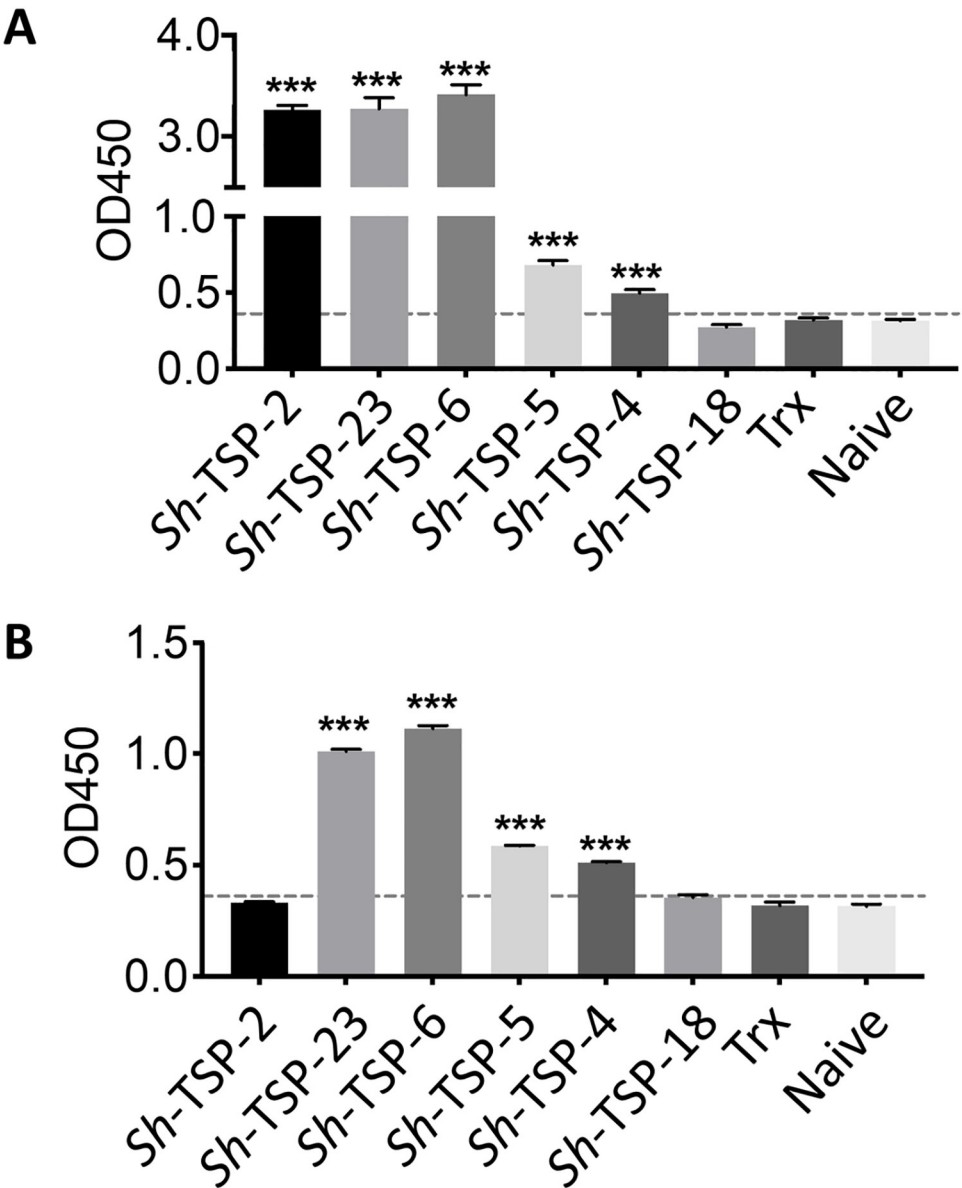

**Fig 4. *Schistosoma haematobium* tetraspanins are recognized by antibodies in the sera of infected mice.** Bar graph showing the detection of *S. haematobium* tetraspanins (TSPs) using *S. haematobium* infected mouse serum (A) and *S. mansoni* infected mouse serum (B). The reactivity cut-off points were determined by comparing the average $OD_{450}$ of infected mouse serum with the average plus 3 standard deviations of naïve mouse serum. A statistical analysis between infected *vs.* naïve mice was performed using a Student's t test. ***P<0.001.

interactions, and some TSPs have been tested as potential vaccines against a range of trematode infections [36, 51–53]. The first TSP identified in *S. mansoni* was *Sm*23 [54] and, since then, other TSPs have been found in the proteome of schistosomes [55–58]. Recently, the *S. haematobium* ortholog of *Sm*23 (*Sh*-TSP-23), and 5 other additional TSPs were identified in a proteomic analysis of the ES products, EVs and tegument from *S. haematobium* adult worms [31] and, in the present study we have characterized these *Sh*-TSPs, and assessed the potential of three of them for diagnosing infection using the urine of naturally infected individuals with different infection intensities.

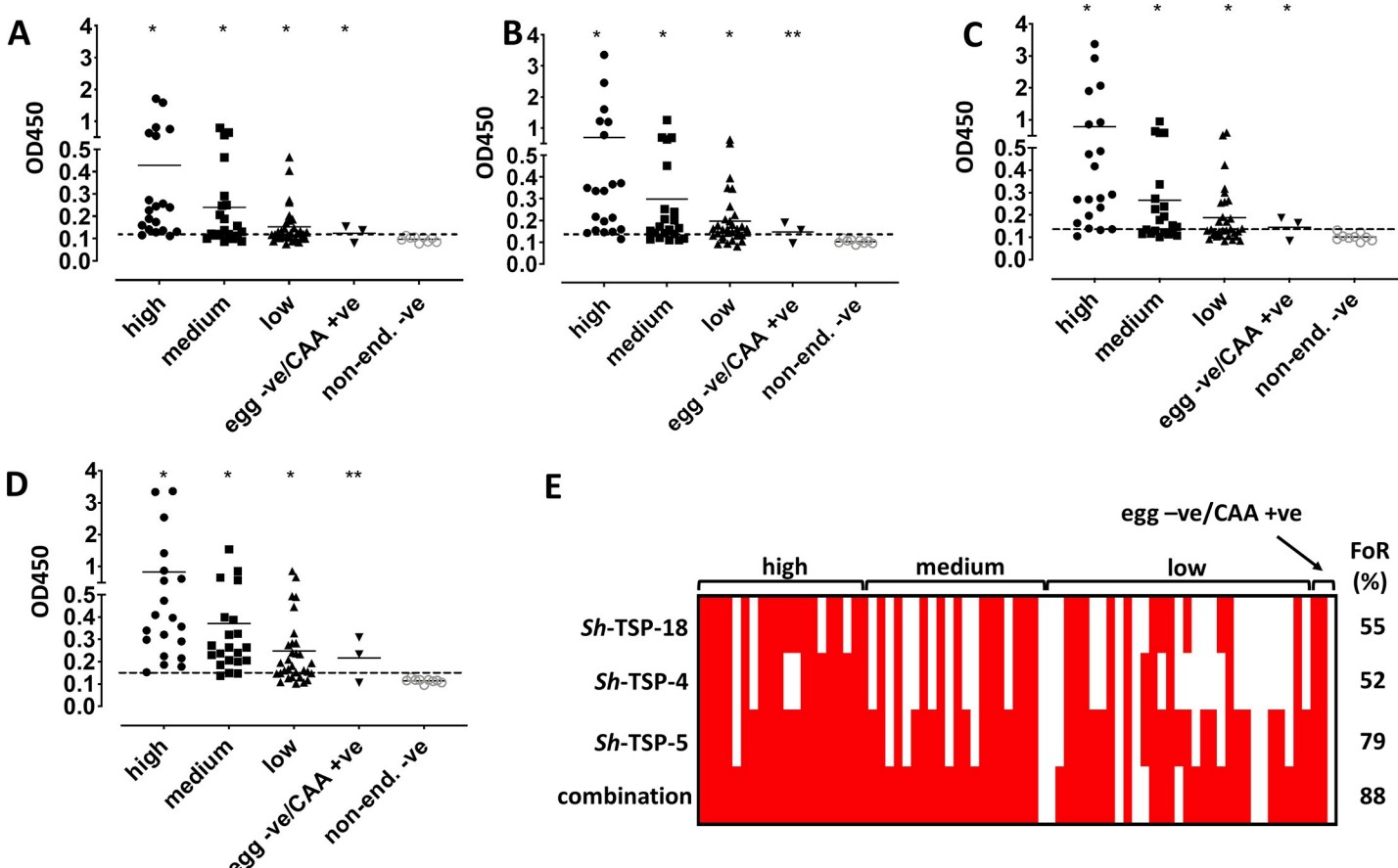

**Fig 5. Recognition of six *Schistosoma haematobium* tetraspanins by urine antibodies from Zimbabwean individuals with different infection intensities.** The antibody level was measured by indirect ELISA and indicated by OD values: *Sh*-TSP-4 (A), *Sh*-TSP-5 (B), *Sh*-TSP-18 (C) and the combination of all three antigens (D). All the data was entered in GraphPad Prism 9 and analysed by Student's t test (infected *vs.* non endemic negative). * *P* < 0.05, ** *P* < 0.01. Urine of non-infected individuals from a non-endemic area was used as negative control. The reactivity cut-off values were determined by adding the average and 3x standard deviations of non-endemic negative individuals (indicated by broken lines). (E) Frequency of recognition (FoR) patterns based on ELISA IgG responses to *E. coli*-expressed recombinant proteins. FoR percentages among the infected populations (sensitivity) are displayed on the right-hand side of the image.

*Sh*-TSP-2, *Sh*-TSP-6 and *Sh*-TSP-23 are grouped together in the CD63 family of TSPs, together with known *S. mansoni* and *S. japonicum* vaccine candidates [36, 52], whereas *Sh*-TSP-4, *Sh*-TSP-5 and *Sh*-TSP-18 are clustered under the uroplakin family of TSPs. All *Sh*-TSPs formed a single clade distinct from vertebrate TSPs. Interestingly, CD63 antigens such as *Sh*-TSP-2 and *Sm*-TSP-2 have been studied as diagnostic markers and vaccine candidates in the past [29, 36]. CD63 and uroplakin members have been suggested to have similar functional properties in *S. japonicum* [59], although the exact roles for uroplakins, either in the infection process or in pathogenesis, is still unkown and further research is needed.

All *Sh-tsp* genes were expressed throughout all the assessed life stages of the parasite, albeit with differing expression patterns, as has been reported for *S. japonicum* [38] and *S. mansoni* [60]. Taken together with their detected presence in tegumental, EV and ES proteomic studies, this data suggests that all *Sh*-TSPs are, upon initial infection, continuously exposed to the immune system. The highest levels of expression for *Sh*-TSP-5, *Sh*-TSP-6, *Sh*-TSP-18, were observed in miracidia, which implies that these *Sh*-TSPs may have specific roles in the intermediate snail host. For *Sh*-TSP-4, *Sh*-TSP-23 and *Sh*-TSP-2, the highest level of expression was observed in cercariae and 24 h schistosomula, respectively. Similarly, the highest expression

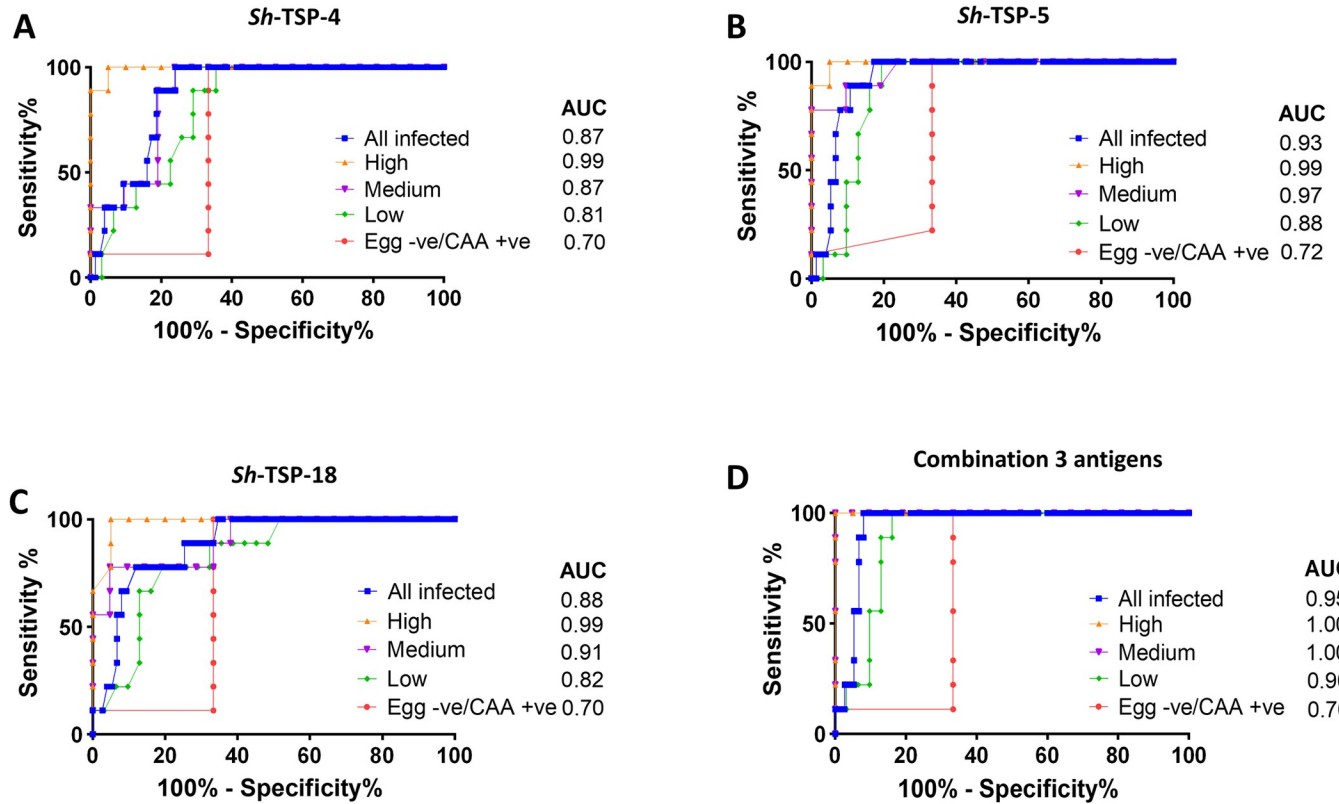

**Fig 6. Receiver operating characteristic (ROC) curves analysis of *Schistosoma haematobium* tetraspanins.** The diagnostic accuracy of *S. haematobium* TSPs to detect antibodies in the urine of infected individuals with differing infection status was measured by the area under the ROC curve (AUC). *Sh*-TSP-4 (A), *Sh*-TSP-5 (B), *Sh*-TSP-18 (C) and the combination of all three antigens (D). Urine of non-infected individuals from non-endemic area was used as negative control.

level of *Sm*-TSP-2 was detected in egg [34]. Furthermore, the highest expression level for the CD63-like TSPs from the human carcinogenic liver fluke, *O. viverrini* (*Ov*-TSP-2 and *Ov*-TSP-3) was observed in the egg stage [35]. These *S. mansoni* and *O. viverrini* TSPs have been shown to be involved in tegument formation, maturation and stability, and thus, homologous *Sh*-TSPs may also be involved in tegument formation and EV secretion [61–64].

Sh-TSP-4, *Sh*-TSP-5, *Sh*-TSP-6 and *Sh*-TSP-23 are located on the tegument surface as well as on various internal organs of adult flukes. Similarly, *Sj*-TSP-5 and *Sj*-TSP-6 are also located on the tegument and internal organs of adult worms whereas *Sj*-TSP-3 *Sj*-TSP-1 are located only in the internal organs of adult worms [59]. In contrast, *Sh*-TSP-2 and *Sh*-TSP-18 are located exclusively on the tegument of adult *S. haematobium* worms, and other TSPs from *S. mansoni*, *O. viverrini* and *S. japonicum* [35, 36, 59, 65] also display this localisation pattern. These tegumental proteins might play an important role in tegument biogenesis and turnover [35, 36, 65]. Since the tegument is the most susceptible structure to host-mediated immune attack [66], the LEL regions of TSPs in trematodes have been tested as vaccine candidates. Immunization of mice with the LEL of two *S. mansoni* TSPs (*Sm*-TSP-1 and *Sm*-TSP-2) significantly decreased adult worm and liver egg burdens [36] and *Sm*-TSP-2 has completed phase I clinical trials [67]. *Sm*23 is one of the independently tested WHO vaccine candidates [68] and its *S. japonicum* ortholog (Sj23) was also found to be an efficacious vaccine in animal models of schistosomiasis [69, 70]. Indeed, *Sm*23 has been shown to be efficacious when delivered as a

DNA vaccine [71], and immunization of mice with a fusion of the *Sj*23 TSP with other vaccine candidates significantly reduced the worm burden and liver eggs in a subsequent *S. japonicum* challenge [69, 70]. Furthermore, antibodies produced against a TSP present in *O. viverrini* EVs blocked the internalization of EVs by cholangiocytes and decreased the production of cytokines that stimulate tumorigenesis [61], while TSPs present in *S. haematobium* EVs elicited protection in a heterologous model of schistosomiasis [30]. These observations demonstrated the usefulness of TSPs in vaccine strategies.

All *Sh*-TSPs except *Sh*-TSP-18 were significantly recognized by serum antibodies from mice experimentally infected with *S. haematobium*. Similarly, TSPs from other platyhelminths such as *T. solium*, *O. viverrini* and *S. japonicum* can be recognized by the antibodies of infected humans and experimental animals [35, 38, 39, 65]. This, together with the localisation studies and gene expression patterns, suggests that these TSPs are accessible to antibodies and are immunogenic during natural infection. Indeed, we have shown *Sh*-TSP2, *Sh*-TSP-6 and *Sh*-TSP-23 to be the targets of IgG responses in the serum and urine of individuals from *S. haematobium*-endemic areas [29]. Interestingly, *Sh*-TSP2 and *Sh*-TSP-18 were not recognized by antibodies from *S. mansoni* infected mice, an observation consistent with the preferential recognition of *Sh*-TSP-2 by *S. haematobium*-infected as opposed to *S. mansoni*-infected individuals [29], implying the potential usefulness of *Sh*-TSP-2 and *Sh*-TSP-18 in the diagnosis of *S. haematobium* infections where both species are co-endemic. Motivated by these results, and despite the low recognition of *Sh*-TSP-18 by the serum of infected mice, we decided to test the utility of *Sh*-TSP-4, *Sh*-TSP-5 and *Sh*-TSP-18 (the diagnostic efficacy of *Sh*-TSP2, *Sh*-TSP-6 and *Sh*-TSP-23 has been reported elsewhere [29]) as biomarkers of *S. haematobium* infection by using them to detect antibodies in the urine of individuals infected with the parasite. Urine, not serum, was selected as the diagnostic fluid due to the relative ease of sample collection, especially in field conditions [72]. In certain infectious disease states, the use of urine over serum for diagnosis may be at the expense of sensitivity, but we posit that, for *S. haematobium* infections, any potential decrease in detection limits may be mitigated by an increased level of IgG present in the urine, relative to that produced by kidney excretion, due to the serum antibody leakage into the bladder caused by egg-induced damage. Each *Sh*-TSP was the target of significantly elevated antibody levels in all cohorts, including egg negative/CAA positive urine samples, compared to uninfected subjects, highlighting the increased diagnostic sensitivity of this approach compared to egg microscopy. Further, the use of *Sh*-TSP combinations in diagnosis increased the predictive accuracy of infection by more comprehensively capturing the breadth of the anti-schistosome antibody response than any single antigen. This is likely explained by the different FoR patterns for each antigen (due to the variation in life-stage expression of each molecule, different antigens may be present in individuals with acute or patent infections) combining to form a "consensus" FoR for the infected population. With regards to diagnostic use, preparations of defined, recombinant antigens can offer an advantage over crude protein mixtures (such as SEA or SWAP) in that they represent a more standardised and sustainable resource for diagnosis. Indeed, the defined antigen preparations described herein have AUC and FoR values which exceed that of SEA [31], making them a potentially effective, as well as rigorous, tool for the non-invasive diagnosis of *S. haematobium* infection.

In the present study, we have fully characterised different *Sh*-TSPs in order to advance the knowledge on this important family of proteins. In addition, this is the first study assessing the diagnostic efficacy of uroplakin-like *Sh*-TSPs, and only the second study describing the use of defined recombinant antigens in the diagnosis of *S. haematobium* infection, and provides the first steps towards the generation of new diagnostic tools against this devastating disease.

## Supporting information

**S1 Fig. Sequence alignment of the open reading frame (ORF) from six *Schistosoma haematobium* tetraspanins and their respective *S. mansoni* homolog.** A multiple sequence alignment was carried out using MUSCLE with defaults and visualised using JalView with default ClustalX colouring.
(TIF)

**S2 Fig. Sequence alignment of the large extracellular loop (LEL) from six *Schistosoma haematobium* tetraspanins and their respective *S. mansoni* homolog.** A multiple sequence alignment was carried out using MUSCLE with defaults and visualised using JalView.
(TIF)

**S3 Fig. Coomassie stained SDS-PAGE and Western blot analysis of *Schistosoma haematobium* tetraspanins.** 1μg of each TSP was electrophoresed on SDS-PAGE and Comassie-stained: Protein ladder (A1), *Sh*-TSP-2 (A2), *Sh*-TSP-4 (A3), *Sh*-TSP-18 (A4), *Sh*-TSP-5 (A5), *Sh*-TSP-23 (A6) and *Sh*-TSP-6 (A7). Western blot of TSPs using an anti-His monoclonal antibody: Protein ladder (B1), *Sh*-TSP-2 (B2), *Sh*-TSP-4 (B3), *Sh*-TSP-18 (B4), *Sh*-TSP-5 (B5), *Sh*-TSP-23 (B6) and *Sh*-TSP-6 (B7). The expected size of recombinant *Sh*-TSP-2 was 12.4 kDa, since it was expressed without a tag. For *Sh*-TSP-4, *Sh*-TSP-5, *Sh*-TSP-6, *Sh*-TSP-18 and *Sh*-TSP-23, since they contained the TRX-tag from the cloning vector, their expected size ranged from ~28 kDa to ~38 kDa.
(TIF)

**S4 Fig. Urine IgG recognition of thioredoxin from Zimbabwean infected individuals with different infection intensities.** The antibody level was measured by indirect ELISA and indicated by OD values. Urine of non-infected individuals from non-endemic area was used as negative control. The reactivity cut-off values were determined by adding the average and 3x standard deviation of non-endemic negative individuals (indicated by broken lines).
(TIF)

**S1 Table. Lists of oligonucleotide primers used for qPCR analysis of *Schistosoma haematobium* tetraspanins.**
(DOCX)

**S2 Table. Lists of sequences from *Bos taurus, Clonorchis sinensis, Danio rerio, Homo sapiens, Mus musculus, Opisthorchis viverrini, Schistosoma haematobium, Schistosoma japonicum* and *Schistosoma mansoni* used for the phylogenetic analysis.**
(DOCX)

**S3 Table. Codes and suggested names for *Schistosoma haematobium* tetraspanins reported. Names have been assigned to match previously identified *Schistosoma mansoni* homologs. In the absence of a previously described homolog, names have been numerically assigned.**
(DOCX)

**S4 Table. Lists of oligonucleotide primers flanking the LEL region of *Schistosoma haematobium* tetraspanins.**
(DOCX)

## Acknowledgments

The authors thank the study participants, as well as the parents/legal guardians and their teachers in Zimbabwe for their support of this study. They are very grateful for the cooperation of

the Ministry of Health and Child Welfare in Zimbabwe. For their technical support, they would like to thank the members of the Department of Biochemistry at the University of Zimbabwe.

## Author Contributions

**Conceptualization:** Mark S. Pearson, Alex Loukas, Javier Sotillo.

**Data curation:** Matt Field, Javier Sotillo.

**Formal analysis:** Gebeyaw G. Mekonnen, Bemnet A. Tedla, Mark S. Pearson, Matt Field, Govert van Dam, Paul L. A. M. Corstjens, Alex Loukas, Javier Sotillo.

**Funding acquisition:** Alex Loukas, Javier Sotillo.

**Investigation:** Gebeyaw G. Mekonnen, Mark S. Pearson, Luke Becker, Govert van Dam, Paul L. A. M. Corstjens, Javier Sotillo.

**Methodology:** Gebeyaw G. Mekonnen, Bemnet A. Tedla, Mark S. Pearson, Luke Becker, Govert van Dam, Paul L. A. M. Corstjens.

**Resources:** Mark S. Pearson, Abena S. Amoah, Takafira Mduluza, Francisca Mutapi, Alex Loukas.

**Software:** Gebeyaw G. Mekonnen, Javier Sotillo.

**Supervision:** Mark S. Pearson, Alex Loukas, Javier Sotillo.

**Visualization:** Matt Field.

**Writing – original draft:** Gebeyaw G. Mekonnen, Mark S. Pearson, Javier Sotillo.

**Writing – review & editing:** Bemnet A. Tedla, Mark S. Pearson, Luke Becker, Abena S. Amoah, Govert van Dam, Paul L. A. M. Corstjens, Francisca Mutapi, Alex Loukas, Javier Sotillo.

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
