## [Decision Letter · Decision Letter 0]

25 Nov 2021

Dear Dr. Sotillo,

Thank you very much for submitting your manuscript "Characterisation of tetraspanins from Schistosoma haematobium and evaluation of their potential as novel diagnostic markers" for consideration at PLOS Neglected Tropical Diseases. As with all papers reviewed by the journal, your manuscript was reviewed by members of the editorial board and by several independent reviewers. The reviewers appreciated the attention to an important topic. Based on the reviews, we are likely to accept this manuscript for publication, providing that you modify the manuscript according to the review recommendations. 

Sincerely,

Bonnie L Webster

Associate Editor

Cinzia Cantacessi

Deputy Editor

Reviewer's Responses to Questions

**Key Review Criteria Required for Acceptance?**

**Methods**

-Are the objectives of the study clearly articulated with a clear testable hypothesis stated?

-Is the study design appropriate to address the stated objectives?

-Is the population clearly described and appropriate for the hypothesis being tested?

-Is the sample size sufficient to ensure adequate power to address the hypothesis being tested?

-Were correct statistical analysis used to support conclusions?

-Are there concerns about ethical or regulatory requirements being met?

Reviewer #1: The authors assessed several Sh tetraspanins for diagnostic value. antibodies were produced that were positive in human urine samples in infected people but not uninfected urinr samples. I had concerns about the number of negative controls but these number are sufficient for the study. Overall the methods and experimental process are appropriate.

Reviewer #2: Accept

Reviewer #3: No specific comments/issues.

Reviewer #4: The methods are clear and detailed. The study design is well thought out and all sections contain an appropriate amount of information/referencing for another to replicate the studies.

**Results**

-Does the analysis presented match the analysis plan?

-Are the results clearly and completely presented?

-Are the figures (Tables, Images) of sufficient quality for clarity?

Reviewer #1: The results of the experiments are clear and well presented. The figures are well done and representative of the data. Organization of the figures, including supplemental figures is good.

Reviewer #2: Accept

Reviewer #3: No specific comments/issues.

Reviewer #4: The results are clearly presented. Data presented in the primary figures of the manuscript seems appropriate, as does the material presented in supplementary files.

**Conclusions**

-Are the conclusions supported by the data presented?

-Are the limitations of analysis clearly described?

-Do the authors discuss how these data can be helpful to advance our understanding of the topic under study?

-Is public health relevance addressed?

Reviewer #1: The conclusions are clear and supported by the data presented. The data presented open possibllities for new diagnostics for Sh infection.

Reviewer #2: Accept

Reviewer #3: No specific comments/issues.

Reviewer #4: The conclusions are aligned well with the data presented.

**Editorial and Data Presentation Modifications?**

Reviewer #1: (No Response)

Reviewer #2: Accept

Reviewer #3: For ease of reading, I would recommend the authors name their new members of the TSP family (ShTSP1, 2 etc) instead of continuing to use the genome ID throughout. They have clearly shown in their supplementary figures that they are TSPs family members. I know this minor change will lead to quite a lot of changes in fugures and manuscript but it was hard to follow at times using the full gene codes.

I would also recommend the authors make it more explicit in the results about their prior publication, as in the results section it is unclear why the three TSPs are chosen to proceed to diagnostics based on the results shown. The prior mouse data didn’t clearly support the decision as MS3_05289 was not recognised nor were they linked in expression or location. I thought originally it was as the were all insoluble in protein production, but the answer mentioned in discussion is that the others were previously already examined. I think some noting of this in the results would aid in clarity. 

Phylogenetics

- methods section didnt seem to convey enough information relating to number of bootstraps, substitution model - it is needed for future understanding of the paper.

- figure legend for figure 1 is too limited - should at least menton the type of phylogenetic programme.

- results coverage may also have noted bootstrap support levels for the clustering as it enhances the arguement

qPCR -method section states the use of 2 delta delta CT but from my understanding this assumes a uniform PCR amplification efficiency of 100%, but I didnt see any proof that this was the case for the transcrpts assessed. 

Finally, I do think some explicit consideration of the polymorphisms highlighted in reference 53 would be important as a potential limatation.

Minor aspects

Table S1 S3 - use the correct prime symbol 

Line 66 – slightly unclear as a “single female adult worm” wouldn’t release those level of eggs as it would surely require the worm to be paired with a male.

L143 – first “ml” needs a space between value and unit.

L436 – reference 58 is in a different form of brackets and is unlikely to be actually reference 58 as this is an S. mansoni paper. Maybe reference 38?

Supplementary Figure 1 and 2 legend - needed a bit more explanation - what do the colours represent?

Supplementary Figure 3 legend- some specifics on expected size of recombinant proteins would have helped.

L881 - spelling of coomassie

Reviewer #4: Line 157: The parentheses around reference 38 appears different from the other references.

Line 163: volume written as ml it more frequently appears in the text as mL, authors should be consistent.

Line 174: no manufacturer is listed for the Superscript III

**Summary and General Comments**

Reviewer #1: This manuscript by Mekonnen et al characterized several tetraspanins from Schistosome haematobium, which were cloned, espressed in bacteria, for which antibodies were produced in mice. Three of these antigens were responsive in Sh infected individuals and may be an option for Sh diagnostics. This makes the study of general interest. The study is exceptionally well done. The methods and experiments are appropriate, clearly explained and properly done.

Reviewer #2: The paper presented by Medonnen et al entitled "Characteristion of tetraspanins from Schistosoma haematobium and evaluation of their potential as a novel diagnostic markers" provides evidence for the use of tetraspanins as new antigens candidates to immunologically diagnose S. haematobium infection.

The paper is very well written and easy to read. The materials and methods are perfectly detailed and clear. The results are in accordance with the methodology used and the discussion is pleasant and easily understood, limiting itself to describe and corroborate their findings with those available in the literature. The figures presented are intuitive and easy to understand. Good for presenting the results. Overall, a very good work that, in my opinion, deserves to be published in PNTDs.

Reviewer #3: Overall an excellent piece of research with appropriate experiments, well displayed and discussed. Great to read more S. haematobium research which is an under-researched parasite given its significance. All authors should be proud.

Reviewer #4: This study presented by Mekonnen et al is a clearly presented and well written account of their work to investigate the potential use of TSPs as a urine-based diagnostic for Schistosoma haematobium. I appreciate the logical approach to presenting this work, first describing the proteins that were selected, outlining the production of recombinant variants, demonstrating their utility in a controlled mouse study before finally testing their utility in human urine samples. Other than a few minor comments, I believe this manuscript to be in good shape. I take no real issue with what is presented or how it is presented. My primary comments (see below) relate to what might be clarified somewhat, or perhaps included in addition to what is currently found in the manuscript.

Comments:

1. In the discussion, the authors mention that MS3_09198 and 01370 have already been shown to be the targets of IgG responses in the serum and urine of S. haematobium positive individuals. Initially, I questioned why these two targets weren't included in the studies presented in this manuscript, as they appear to elicit the most robust responses in figure 4. As it is currently written in this manuscript, it's not really clear why MS3_09198 and 01370 are included since they aren't investigated in the human urine or AUC analysis.

2. After reading the results I left wondering whether there was a relationship between the actual egg counts in the human urine samples and the ELISA OD values. I'm not sure if this information is available beyond the more general high, medium and low classifications used, but it would be interesting (if possible) to compare ELISA values to actual egg counts and to directly compare ELISA values between the three tests. Do the urine samples with the highest egg counts result in the highest ELISA values and is this consistent across the three targets?

3. A nitpicking detail related to Figure 5. Panel B is the only ELISA that uses a different scale than the other panels. For comparison purposes it would be helpful to unify the Y-axis of panel B to match the others.

4. I may have missed it, but I didnt come across a reference to Figure 5E in the results section.

PLOS authors have the option to publish the peer review history of their article (what does this mean?). If published, this will include your full peer review and any attached files.

Reviewer #1: No

Reviewer #2: No

Reviewer #3: No

Reviewer #4: No

Figure Files:

Data Requirements:

Reproducibility:

References

---

## [Editor Report · Decision Letter 1]

6 Jan 2022

Dear Dr. Sotillo,

We are pleased to inform you that your manuscript 'Characterisation of tetraspanins from Schistosoma haematobium and evaluation of their potential as novel diagnostic markers' has been provisionally accepted for publication in PLOS Neglected Tropical Diseases.

Best regards,

Bonnie L Webster

Associate Editor

Cinzia Cantacessi

Deputy Editor

---

## [Editor Report · Acceptance letter]

18 Jan 2022

Dear Dr. Sotillo,

We are delighted to inform you that your manuscript, "Characterisation of tetraspanins from <i>Schistosoma haematobium<i> and evaluation of their potential as novel diagnostic markers," has been formally accepted for publication in PLOS Neglected Tropical Diseases.

Best regards,

Shaden Kamhawi

co-Editor-in-Chief

Paul Brindley

co-Editor-in-Chief
